# Design of a DSP-Based Motion-Cueing Algorithm Using the Kinematic Solution for the 6-DoF Motion Platform

**Ming-Yen Wei** 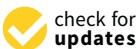

Aeronautical System Research Division, Simulation System Section, National Chung-Shan Institute of Science and Technology, Taichung 407, Taiwan; b515410@ncsist.org.tw; Tel.: +886-4-2702-3051

**Abstract:** A motion-cueing algorithm is a motion simulation system that makes the pilot feel the flight motion by calculating the attitude of the platform. This paper presents the design a kinematics model and two motion-cueing algorithms for a multi-axis motion platform. Firstly, the relationship between each axis is derived from the kinematics theory and motion platform transformation. Next, two motion-cueing algorithms are designed providing the pilot with the bodily sensations of the 6-DoF motion platform. By using a hardware-in-the-loop (HIL) approach simulated in a real-time digital simulator, the control operations are performed in a digital signal processor (DSP). All of the motion-cueing algorithms, including the classical washout algorithm and the optimal control algorithm, are realized through a DSP, TMS-320F-28377D. The simulation results verify the theoretical analysis and illustrate the correctness and practicability of the proposed method.

**Keywords:** kinematics; motion-cueing algorithm; digital signal processor (DSP); real-time control



## 1. Introduction

The training of pilots worldwide is typically lengthy. Therefore, flight training simulators are in high demand and serve as essential tools in flight training [1]. As indicated in [2], a complete high-fidelity motion simulation system was first constructed in the 1950s. Subsequently, in 1965, Stewart from the United Kingdom proposed a Stewart platform with six degrees of freedom (DoF) for flight simulators based on a parallel mechanism [3]. Because of the high research potential of the Stewart platform, many parallel mechanism-related applications have emerged since the 1980s, such as vehicle simulation systems [4], precision positioning [5], active antivibration platforms [6], adjustment platforms for astronomical telescopes [7], and virtual reality motion platforms [8].

Current platforms, such as the six-degrees-of-freedom Stewart platform and the six-degrees-of-freedom crank arm platform, have a movement space with six degrees of freedom. The upper platform can be incorporated with a cockpit and used for the simulation of real vehicles [9,10]. Due to the simple control, high degree of freedom, high efficiency, and high carrying capacity (tons) characteristics of the Stewart platform, the production cost of linear actuators is high, and the platform height to be positioned in the center is too high [11]. Therefore, there is more and more research on the crank arm platform. The reason to have comparable high-fidelity motion, which reduces the manufacturing cost, is to enable its use in professional training simulators and entertainment simulators [12–14]. However, with the rapid advances in science and technology in recent years, and the structure, performance, and power of modern military products often exceed the movement limits of the previous Stewart platform. In [15], the KUKA robotic arm was used as a motion platform, and inverse kinematics were applied to derive the six degrees of freedom of a tandem mechanism movement, after which the classical washout algorithm was employed to implement a racing simulator. In addition, Fan investigated a human centrifuge [16]. However, the human centrifuges that have been developed have three degrees of freedom and provide G-forces for fixed training and pursuit training and establish a high-fidelity

pilot module that considers both physical fidelity and function, thus providing trainers with a real experience. Berthoz et al. [17] investigated different motion scale factors in a driving simulator. Through motion feedback, participants drove more carefully and had better control of the car. Therefore, they could better anticipate the car's dynamic behavior and were not as surprised when the car did crash. Very reduced or absent motion cues significantly degrade driving performance.

The simulator mainly applies the wash-out algorithm for the balance organ of the human body, that is, the vestibular system of the inner ear. In addition, the trainer is provided with an immersive experience through changes in the position, velocity, and acceleration of the motion platform which improves the overall efficiency of the trainer. The sensitive range of motion is shown on the motion platform [18]. The motion platform operates on the principle of using the inner ear vestibular system to perceive a considerable level of linear and rotational speed and acceleration to achieve the desired feeling of simulation. Therefore, motion platforms are applied to simulate actual vehicle movements to create realistic forward and backward acceleration and road vibration that occurs during car movement in order to achieve the effect of simulator training [19]. In addition, to improve the control of six-servo synchronized tracking, in [20], an adaptive control technology–integrated motion-cueing algorithm was proposed for adaptive parameter estimation at a limited interval to obtain the amount of control required by the control platform. In [21], a wave filter model was designed using the optimization theory, and the optimal parameters were determined using the recursive genetic algorithm. In [22], a fuzzy control rule base was established for the rate limiter within the washout algorithm, and a rate limiter with dynamic adjustment was designed using the specific force error. In [23], the predictive control theory served as the basis for designing a filter model, which was used to reference the parameters required for each modification of the wave filter.

In this paper, two motion-cueing algorithms using software-embedded applications are presented. The DSP-based algorithm design is analyzed by deriving the mathematical relationship of the motion platform using forward and inverse kinematics. Because the design adopted a real time digital simulator and improved the number of calculations performed, the algorithms in real time can be reduced. To the best of our knowledge, the ideas we mentioned have not been presented in any previously published papers [1–23]. As a result, this paper presents some new ideas on the realization of motion-cueing algorithms for a 6-DoF motion platform, including a classical washout algorithm and an optimized tracking motion-cueing algorithm.

## 2. Kinematics of the Six-DoF Motion Platform

In this section, the 6DoF forward kinematics and inverse kinematics are established. In Figure 1a–c the motion platform considered in this study is presented. Figure 1a shows the co-ordinate system designed for the motion platform in this paper. The architecture is based on six completely decoupled degrees of freedom. Therefore, each degree of freedom can be controlled independently, as shown in Figure 1b. The 6DoF platform includes three rack-and-pinion electric linear actuators and three electric rotary actuators that provides the pilots with kinesthetic sensations related to motion. Two servo bilateral drives of the X1 and X2 axes are the surge axis. The Y axis is the sway axis. The heave axis uses the two servo bilateral drives of the Z1 and Z2 axes. All power and electrical signals are transmitted from the yaw axis to the pitch axis through three electric slip rings, as well as being used in the platform cockpit. The rotation part includes a roll axis servo drive of the A axis, a pitch axis servo drive of the B axis, and a yaw axis servo drive of the C axis. Figure 1c shows the parts rotated from the inside to the outside by the cockpit, which are the pitch axis, roll axis and yaw axis.

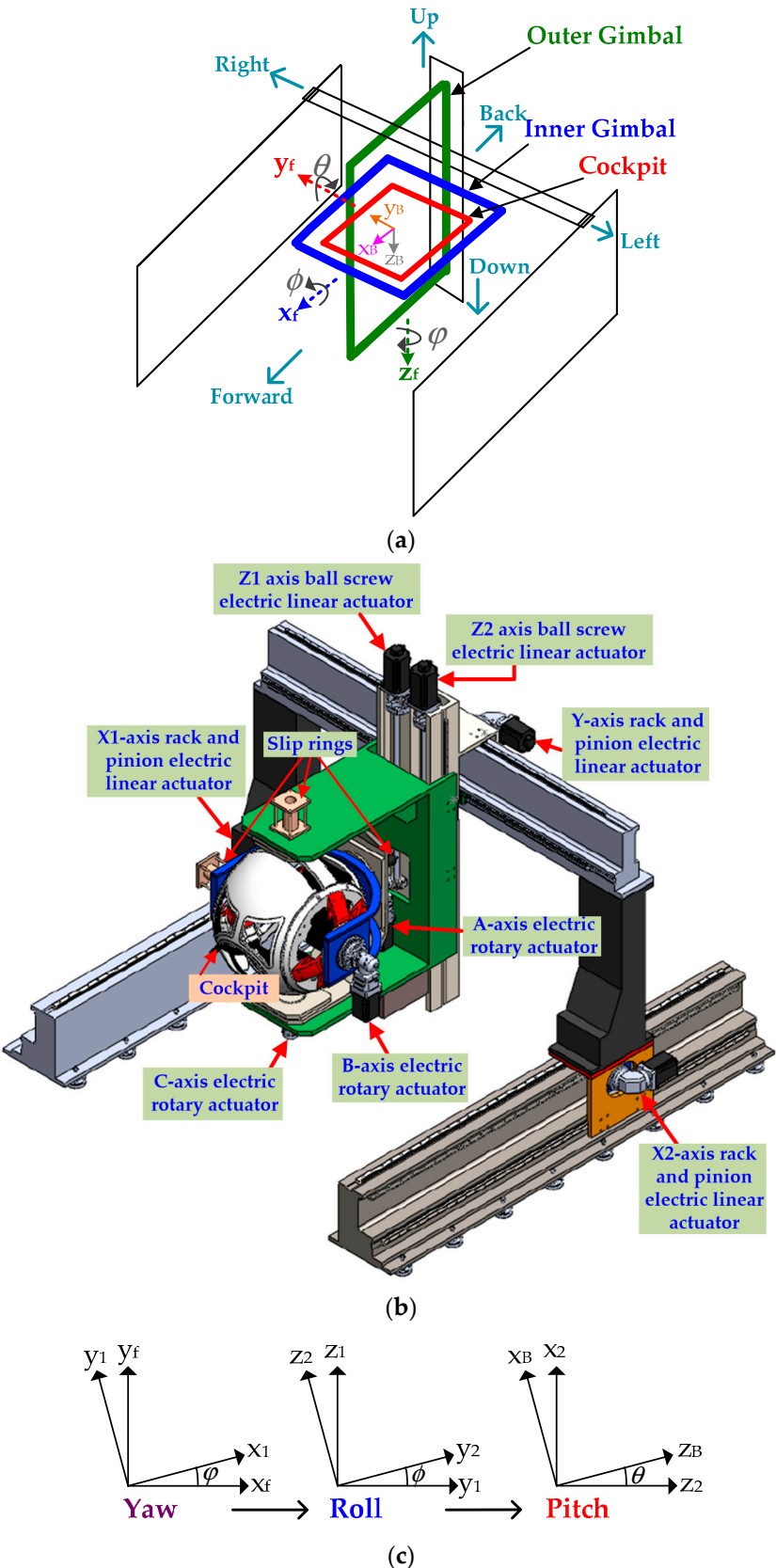

**Figure 1.** Six-DoF platform: (**a**) configuration of the co-ordinate system, (**b**) 3D design drawing, and (**c**) rotation relationship.

The co-ordinate conversion relationship is:

$$
\begin{bmatrix} x_f \\ y_f \\ z_f \end{bmatrix} = \begin{bmatrix} \cos\theta & 0 & \sin\theta \\ 0 & 1 & 0 \\ -\sin\theta & 0 & \cos\theta \end{bmatrix} \begin{bmatrix} x_1 \\ y_1 \\ z_1 \end{bmatrix} \tag{1}
$$

$$
\begin{bmatrix} x_1 \\ y_1 \\ z_1 \end{bmatrix} = \begin{bmatrix} 1 & 0 & 0 \\ 0 & \cos\phi & -\sin\phi \\ 0 & \sin\phi & \cos\phi \end{bmatrix} \begin{bmatrix} x_2 \\ y_2 \\ z_2 \end{bmatrix} \tag{2}
$$

$$
\begin{bmatrix} x_2 \\ y_2 \\ z_2 \end{bmatrix} = \begin{bmatrix} \cos\psi & -\sin\psi & 0 \\ \sin\psi & \cos\psi & 0 \\ 0 & 0 & 1 \end{bmatrix} \begin{bmatrix} x_B \\ y_B \\ z_B \end{bmatrix} \tag{3}
$$

*2.1. Forward Kinematics*

According to the conversion relationship in Equations (1)–(3), the relationship between the inertial co-ordinate vector $V^I$ and the body co-ordinate vector $V^B$ is given by:

$$
V^I = {}^IR^B V^B \tag{4}
$$

where ${}^IR^B$ is the rotation matrix from the body co-ordinates to the inertial co-ordinates. It can be written as:

$$
{}^IR^B = R_y R_x R_z \tag{5}
$$

where $R_x$, $R_y$, and $R_z$ are the rotation matrices on the $x$ axis, $y$ axis, and $z$ axis, respectively. Substituting Equation (5) into Equations (1)–(3), can be expressed as:

$$
\begin{aligned}
{}^IR^B &= \begin{bmatrix} \cos\theta & 0 & \sin\theta \\ 0 & 1 & 0 \\ -\sin\theta & 0 & \cos\theta \end{bmatrix} \begin{bmatrix} 1 & 0 & 0 \\ 0 & \cos\phi & -\sin\phi \\ 0 & \sin\phi & \cos\phi \end{bmatrix} \begin{bmatrix} \cos\psi & -\sin\psi & 0 \\ \sin\psi & \cos\psi & 0 \\ 0 & 0 & 1 \end{bmatrix} \\
&= \begin{bmatrix} \cos\psi\cos\theta + \sin\phi\sin\psi\sin\theta & \cos\psi\sin\phi\sin\theta - \cos\theta\sin\psi & \cos\phi\sin\theta \\ \cos\phi\sin\varphi & \cos\phi\cos\psi & -\sin\phi \\ \cos\theta\sin\phi\sin\psi - \cos\psi\sin\theta & \sin\varphi\sin\theta + \cos\varphi\cos\theta\sin\phi & \cos\phi\cos\theta \end{bmatrix}
\end{aligned} \tag{6}
$$

When the platform is in translational motion, the conversion matrix ${}^IT^B$ from the body co-ordinates to the inertial co-ordinates is:

$$
\begin{aligned}
{}^IT^B &= \begin{bmatrix} \cos\psi\cos\theta + \sin\phi\sin\psi\sin\theta & \cos\psi\sin\phi\sin\theta - \cos\theta\sin\psi & \cos\phi\sin\theta & x \\ \cos\phi\sin\psi & \cos\phi\cos\psi & -\sin\phi & y \\ \cos\theta\sin\phi\sin\psi - \cos\psi\sin\theta & \sin\psi\sin\theta + \cos\psi\cos\theta\sin\phi & \cos\phi\cos\theta & z \\ 0 & 0 & 0 & 1 \end{bmatrix} \\
&= \begin{bmatrix} n_x & o_x & a_x & p_x \\ n_y & o_y & a_y & p_y \\ n_z & o_z & a_z & p_z \\ 0 & 0 & 0 & 1 \end{bmatrix}
\end{aligned} \tag{7}
$$

The aforementioned equation is the forward kinematics model of the platform. From the platform axis co-ordinates $(x, y, z, \phi, \theta, \psi)$, attitude $(n_x, n_y, n_z)$ in $x$ direction, attitude $(o_x, o_y, o_z)$ in $y$ direction, and attitude $(a_x, a_y, a_z)$ in $z$ direction, the position $(p_x, p_y, p_z)$ of the body co-ordinates is obtained.

The rotational angular velocity of the body co-ordinates can be expressed as:

$$
\vec{\omega}^B = \begin{bmatrix} p \\ q \\ r \end{bmatrix} \tag{8}
$$

The rotational angular velocity of the axis co-ordinates can be expressed as:

$$\dot{\Phi} = \begin{bmatrix} \dot{\phi} \\ \dot{\theta} \\ \dot{\psi} \end{bmatrix} \tag{9}$$

As observed in Figure 1c, Equations (8) and (9) are arranged as:

$$p\vec{x}_B + q\vec{y}_B + r\vec{z}_B = \dot{\theta}\vec{y}_1 + \dot{\phi}\vec{x}_2 + \dot{\psi}\vec{z}_B \tag{10}$$

Equations (2) and (3) are arranged as:

$$\begin{aligned} \vec{y}_1 &= \cos\phi\,\vec{y}_2 - \sin\phi\,\vec{z}_2 \\ &= (\cos\phi\sin\psi)\,\vec{x}_B + (\cos\phi\cos\psi)\,\vec{y}_B - \sin\phi\,\vec{z}_B \end{aligned} \tag{11}$$

By substituting Equations (3) and (11) into Equation (10), it is not difficult to obtain:

$$\begin{aligned} &p\vec{x}_B + q\vec{y}_B + r\vec{z}_B \\ &= \dot{\theta}\left[(\cos\phi\sin\psi)\,\vec{x}_B + (\cos\phi\cos\psi)\,\vec{y}_B - \sin\phi\,\vec{z}_B\right] + \dot{\phi}\left(\cos\psi\,\vec{x}_B - \sin\psi\,\vec{y}_B\right) + \dot{\psi}\vec{z}_B \\ &= \left(\dot{\theta}\cos\phi\cos\psi + \dot{\phi}\cos\psi\right)\vec{x}_B + \left(\dot{\theta}\cos\phi\cos\psi - \dot{\phi}\sin\psi\right) + \left(\dot{\psi} - \dot{\theta}\sin\phi\right)\vec{z}_B \end{aligned} \tag{12}$$

Then, the rotational angular velocity of the axis co-ordinates with respect to the body co-ordinates can be expressed as:

$$\begin{bmatrix} p \\ q \\ r \end{bmatrix} = \begin{bmatrix} \cos\psi & \cos\phi\cos\psi & 0 \\ -\sin\psi & \cos\phi\cos\psi & 0 \\ 0 & -\sin\phi & 1 \end{bmatrix} \begin{bmatrix} \dot{\phi} \\ \dot{\theta} \\ \dot{\psi} \end{bmatrix} \tag{13}$$

### 2.2. Inverse Kinematics

It is possible to calculate the required platform axis co-ordinates $(x, y, z, \phi, \theta, \psi)$, x-direction attitude $\vec{n}$, y-direction attitude $\vec{o}$, z-direction attitude $\vec{a}$, and the position $(p_x, p_y, p_z)$. First, Equation (5) can be rewritten as:

$$\begin{aligned} R_y^{-1I}R^B &= R_x R_z \\ &= \begin{bmatrix} \cos\psi & -\sin\psi & 0 \\ \cos\phi\sin\psi & \cos\phi\cos\psi & -\sin\phi \\ \sin\phi\sin\psi & \cos\psi\sin\phi & \cos\phi \end{bmatrix} \end{aligned} \tag{14}$$

In addition, Equation (7) includes the attitudes $\vec{n}$, $\vec{o}$, and $\vec{a}$ of the x-direction, y-direction, and z-direction, respectively, as follows:

$$\begin{aligned} &\begin{bmatrix} n_x\cos\theta - n_z\sin\theta & o_x n_x\cos\theta - o_z\sin\theta & a_x\cos\theta - a_z\sin\theta \\ n_y & o_y & a_y \\ n_x\sin\theta + n_z\cos\theta & o_x\sin\theta + o_z\cos\theta & a_x\sin\theta + a_z\cos\theta \end{bmatrix} \\ &= \begin{bmatrix} \cos\psi & -\sin\psi & 0 \\ \cos\phi\sin\psi & \cos\phi\cos\psi & -\sin\phi \\ \sin\phi\sin\psi & \cos\psi\sin\phi & \cos\phi \end{bmatrix} \end{aligned} \tag{15}$$

From Equation (15), the following is obtained:

$$a_x\cos\theta - a_z\sin\theta = 0 \tag{16}$$

where:

$$\tan\theta = \frac{\sin\theta}{\cos\theta} = \frac{a_x}{a_z} \tag{17}$$

The rotation angle of the pitch axis can be obtained as follows:

$$\theta = \tan^{-1}\left(\frac{a_x}{a_z}\right) \tag{18}$$

From Equation (15), we can observe that when $a_x = a_z = 0$, $\cos\phi = 0$ means $\phi = \pm 90°$. Figure 1a shows that the inner gimbal and outer gimbal are coaxial. After obtaining $\theta$ from Equation (18) by using the same method it can be expressed as:

$$\sin\phi = -(a_y) \tag{19}$$

$$\cos\phi = (a_x \sin\theta + a_z \cos\theta) \tag{20}$$

$$\sin\psi = -(o_x n_x \cos\theta - o_z \sin\theta) \tag{21}$$

$$\cos\psi = (n_x \cos\theta - n_z \sin\theta) \tag{22}$$

The angle of the roll axis can be obtained from Equations (19) and (20):

$$\tan\phi = \frac{\sin\phi}{\cos\phi} = -\left(\frac{a_y}{a_x \sin\theta + a_z \cos\theta}\right) \tag{23}$$

The yaw axis angle is expressed as:

$$\tan\psi = \frac{\sin\psi}{\cos\psi} = \left(\frac{-o_x n_x \cos\theta + o_z \sin\theta}{n_x \cos\theta - n_z \sin\theta}\right) \tag{24}$$

The arctangent function can be obtained from Equations (23) and (24):

$$\phi = \tan^{-1}\left(\frac{-a_y}{a_x \sin\theta + a_z \cos\theta}\right) \tag{25}$$

$$\psi = \tan^{-1}\left(\frac{-o_x n_x \cos\theta + o_z \sin\theta}{n_x \cos\theta - n_z \sin\theta}\right) \tag{26}$$

The three axis co-ordinates of the platform are obtained by Equations (18), (25) and (26), and the three surge axis, sway axis, and heave axis relationships are expressed by Equation (7) as:

$$x = p_x, \; y = p_y, \; z = p_z \tag{27}$$

Finally, through the above derivation process of the kinematics model, Equations (18) and (25)–(27), which are required by the axis co-ordinates, are determined.

## 3. Motion-Cueing Algorithm

To fully realize an actual motion feeling using the simulation platform, understanding what the human body feels in the actual simulation is necessary so that pilots can achieve similar analog signals on the simulation platform. Therefore, the operating mechanism of the human body's perception system must be understood. In motion simulation, the perception of visual-effect movement mainly involves the change in the image and display of the instrument in the vehicle. Motion simulation helps realize the motion perceived by humans with a high sensitivity through platform movement, enhancing the effect of virtual reality.

### 3.1. Classical Washout Algorithm Design

The motion-cueing algorithm was primarily used to create an infinite space of acceleration and angular velocity in a limited motion space. The specific force of the nongravitational force in the body co-ordinate system can be expressed as:

$$\vec{f}_s^B = {}^B R^I \vec{a}_s^I - {}^B R^I \vec{\delta}^I \tag{28}$$

where:

$$\vec{f}_s^B = \begin{bmatrix} f_{sx}^B \\ f_{sy}^B \\ f_{sz}^B \end{bmatrix} \tag{29}$$

$$^B R^I = \left( {}^I R^B \right)^{-1} = \begin{bmatrix} \cos\psi\cos\theta + \sin\phi\sin\psi\sin\theta & \cos\phi\sin\psi & \cos\theta\sin\phi\sin\psi - \cos\psi\sin\theta \\ \cos\psi\sin\phi\sin\theta - \cos\theta\sin\psi & \cos\phi\cos\psi & \sin\psi\sin\theta + \cos\psi\cos\theta\sin\phi \\ \cos\phi\cos\theta & -\sin\phi & \cos\phi\cos\theta \end{bmatrix} \tag{30}$$

$$\vec{\delta}^I = \begin{bmatrix} 0 \\ 0 \\ g \end{bmatrix} = \begin{bmatrix} 0 \\ 0 \\ 9.8 \end{bmatrix} \tag{31}$$

$$\vec{a}_s^I = \begin{bmatrix} a_{sx}^I \\ a_{sy}^I \\ a_{sz}^I \end{bmatrix} \tag{32}$$

where $^B R^I$ is the inverse matrix of Equation (6) and is the rotation matrix converted from inertial co-ordinates to body co-ordinates, $\vec{\delta}^I$ is the gravitational acceleration, and $\vec{a}_s^I$ denotes the acceleration of the inertial co-ordinate motion. Considering the rotation matrix, Equation (28) can be expressed as:

$$\vec{f}_s^B = \vec{a}_s^B - {}^B R^I \vec{\delta}^I \tag{33}$$

where $\vec{a}_s^B$ is the acceleration of the body co-ordinate motion. In the case of a small rotation angle, Equation (30) can be written as:

$$^B R^I = \begin{bmatrix} 1 & \psi & -\theta \\ -\psi & 1 & \phi \\ \theta & -\phi & 1 \end{bmatrix} \tag{34}$$

Then, Equation (33) is written as:

$$\vec{f}_s^B = \vec{a}_s^B + g \begin{bmatrix} \theta \\ -\phi \\ -1 \end{bmatrix} \tag{35}$$

The combination of the gravitational acceleration and the acceleration $\vec{a}_s^B$ of the body co-ordinate motion can yield the actual simulator cockpit acceleration $\vec{f}_s^B$. Because of the limited motion space of the simulator and high-frequency acceleration, $\vec{a}_s^B$ can only be used to generate a high-frequency $\vec{f}_s^B$. Low-frequency or continuous $\vec{f}_s^B$ can be generated by titling the simulator at an angle by using the second term on the right-hand side in Equation (35). The motion-cueing algorithm can help separate the high-frequency and low-frequency of $\vec{f}_s^B$ and generate these frequencies using different motion actions of the simulator. In terms of rotation, Equations (8) and (9) were used to derive the rotational angular velocity of body co-ordinates $\vec{\omega}_s^B$ of the simulator and the rotational angular

velocity of the axis co-ordinate $\dot{\Phi}_s$, with the subscript s indicating the physical quantity of the simulator. Except when performing aerobatics during the flight of an aircraft, yaw is not constrained during normal flight because it enables the pilot to make stable turns, resulting in a limited roll and pitch range. Thus, a high-pass filter is required to filter out low-frequency signals with excessively large rotation angles, after which the high-frequency angular velocity generates a simulation of rotation. Figure 2 presents the block diagram of the motion-cueing algorithm. According to the figure, the classical washout algorithm was divided into upper and lower sections after scaling. Furthermore, the upper section was divided into high- and low-frequency signals. After the high-frequency signal was filtered out by the high-pass filter, the motion acceleration required by the platform was generated. The low-frequency signal was used to generate the inclination angle required by the platform.

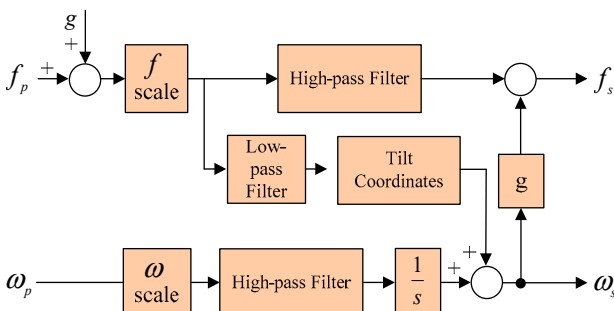

**Figure 2.** Block diagram of classical washout algorithm.

### 3.2. Optimal Control Algorithm Design

In order to truly present the specific force of the cockpit and the rotational angular velocity of the simulator cockpit the optimal control motion-cueing algorithm must be designed. The input-output relationship of the simulator is shown in Figure 3. The input to the simulator is the angular velocity $\omega_p$ and specific force $f_p$ of the vehicle, while the output is the angular velocity $\omega_s$ and specific force $f_s$ generated by the simulator motion. The algorithm yields $\omega_s = \omega_p$ and $f_s = f_p$ from the simulator in the limited motion space.

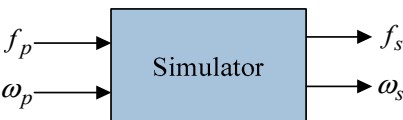

**Figure 3.** The input-output relationship of the algorithm.

In Figure 4 shows the model of the simulator. The equation for the state of the simulator must be established first.

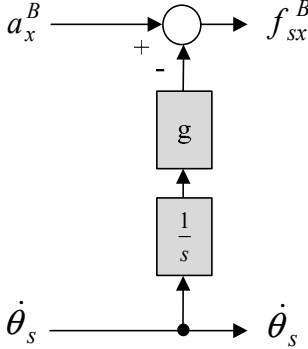

**Figure 4.** The model of the simulator.

As Equation (35) did not have a rotation angle $\psi$, the heave and yaw motions were mutually independent with no coupling term for acceleration and velocity between them. In addition, the first column of the matrix Equation (35) was similar to the second column, therefore, the surge and pitch motions of the equation in the first column of the equation were considered. The sway and roll motions of the equation in the second column have similar resulting motions as those in the first column. The state equation can be expressed as [24]:

$$\dot{X} = AX + BU$$
$$Y = CX + DU \tag{36}$$

The relative symbols are described as:

$$A = \begin{bmatrix} 0 & 1 & 0 \\ 0 & 0 & 0 \\ 0 & 0 & 0 \end{bmatrix}, \ B = \begin{bmatrix} 0 & 0 \\ 1 & 0 \\ 0 & 1 \end{bmatrix}, \ C = \begin{bmatrix} 0 & 0 & g \\ 0 & 0 & 0 \end{bmatrix}, \ D = \begin{bmatrix} 1 & 0 \\ 0 & 1 \end{bmatrix} \tag{37}$$

where $A$, $B$, $C$, $D$ are the parameters of the state-space representation, $X = \begin{bmatrix} x_1 & x_2 & x_3 \end{bmatrix}^T$ denotes the state vector, $x_1$ denotes the displacement, $x_2$ denotes the velocity, $x_3$ denotes the rotation angle, $U = \begin{bmatrix} a_x^B & \dot{\theta}_s \end{bmatrix}^T$ is the input vector, and $Y = \begin{bmatrix} f_{sx}^B & \dot{\theta}_s \end{bmatrix}^T$ is the output vector. The optimal theory in this study was based on [25] and assumes that the angular velocity and acceleration of the vehicle were random signals with a first-order rational function spectrum. The mathematical model is:

$$\dot{Z} = \begin{bmatrix} -\beta_1 & 0 \\ 0 & -\beta_2 \end{bmatrix} Z + \begin{bmatrix} \beta_1 & 0 \\ 0 & \beta_2 \end{bmatrix} W$$
$$= A_n Z + B_n W \tag{38}$$

$$\widetilde{Y} = \begin{bmatrix} 1 & 0 \\ 0 & 1 \end{bmatrix} Z = C_n Z \tag{39}$$

where $\widetilde{Y} = \begin{bmatrix} f_{px} & \dot{\theta}_p \end{bmatrix}^T$ is the specific force and rotational angular velocity of the simulator cockpit and $W$ denotes the Gaussian white noise. Thus, according to Equations (36) and (39), the tracking error can be defined as:

$$e = Y - \widetilde{Y}. \tag{40}$$

Then, the cost function can be designed as:

$$V = \lim_{t \to \infty} \frac{1}{T} E \left[ \int_0^T \left( e^T Q_1 e + X^T Q_2 X + U^T R_1 U \right) dt \right] \tag{41}$$

where $R_1$ is a positive definite matrix and $Q_1$ and $Q_2$ are positive semi-definite matrices. The first term was used to penalize the input signal to avoid controller saturation, and the acceleration and angular velocity generated by the simulator could be reduced by increasing $R_1$. The second term in the integral formula was used to penalize the tracking error $e$, and increasing $Q_1$ can yield a more favorable simulation fidelity. The third term was used to penalize the motion space of the simulator, and the simulator's motion range could be reduced by increasing $Q_2$.

Considering Equations (36), (38) and (39), let $\overline{X} = \begin{bmatrix} X & Z \end{bmatrix}^T$ as follows:

$$\dot{\overline{X}} = \begin{bmatrix} A & 0_{3\times2} \\ 0_{2\times3} & A_n \end{bmatrix} \overline{X} + \begin{bmatrix} B \\ 0_{2\times2} \end{bmatrix} U + \begin{bmatrix} 0_{3\times2} \\ B_n \end{bmatrix} W$$
$$= \overline{A}\overline{X} + \overline{B}U + \overline{\Gamma}W \tag{42}$$

$$\begin{bmatrix} e \\ X \end{bmatrix} = \begin{bmatrix} C & -C_n \\ I_{3\times3} & 0_{3\times2} \end{bmatrix} \overline{X} + \begin{bmatrix} D \\ 0_{3\times2} \end{bmatrix} U \tag{43}$$
$$= \overline{C}\overline{X} + \overline{D}U$$

The three terms in the integral equation of (41) were then substituted into Equations (42) and (43) to yield the following:

$$
\begin{aligned}
& e^T Q_1 e + X^T Q_2 X + U^T R_1 U \\
&= \begin{bmatrix} e^T & X^T \end{bmatrix} \begin{bmatrix} Q_1 & 0_{2\times3} \\ 0_{3\times2} & Q_2 \end{bmatrix} \begin{bmatrix} e \\ X \end{bmatrix} + U^T R_1 U \\
&= \overline{X}^T \overline{C}^T diag\{Q_1, Q_2\} \overline{C}\overline{X} + 2\overline{X}^T \overline{C}^T diag\{Q_1, Q_2\} \overline{D}U + U^T (R_1 + D^T Q_1 D) U \\
&= \overline{X}^T Q\overline{X} + 2\overline{X}^T R_{12} U + U^T R U
\end{aligned}
\tag{44}
$$

where *diag* denotes the diagonal square matrix. Substituting it into Equation (41) can be expressed as:

$$V = \lim_{t\to\infty} \frac{1}{T} E\left[ \int_0^T \left( {}^T\overline{X}^T Q\overline{X} + 2\overline{X}^T R_{12} U + U^T R U \right) dt \right] \tag{45}$$

By solving $K_1$ and $K_2$, the cost function can be minimized. The optimal controller is expressed as follows:

$$U = -\begin{bmatrix} K_1 & K_2 \end{bmatrix} \begin{bmatrix} X \\ Z \end{bmatrix} \tag{46}$$
$$= -K_1 X - K_2 Z$$

Substituting Equation (39) into the Equation (46) yields:

$$U = -K_1 X - K_2 C_n^{-1} \widetilde{Y} \tag{47}$$

Next, the Hamiltonian function is defined as follows:

$$H = \overline{X}^T Q\overline{X} + 2\overline{X}^T R_{12} U + +2U^T R U + \lambda^T \left( A\overline{X} + BU + \Gamma W \right) \tag{48}$$

to satisfy the condition of $\partial H / \partial U = 0$, which yields:

$$U = -\frac{1}{2} R^{-1} B^T \lambda - R^{-1} \overline{X}^T R_{12} \tag{49}$$

the Hamiltonian–Jacobi equation yields:

$$-\frac{\partial J^*}{\partial t} = H(\overline{X}, \lambda^*, U^*, t) = H(\overline{X}, \frac{\partial J^*}{\partial \overline{X}}, U^*, t) \tag{50}$$

The PDE boundary conditions of (50) is as follows:

$$\left. \frac{\partial J^*}{\partial \overline{X}} \right|_{t=T} = \lambda^*(T) = 2P(T)\overline{X} \tag{51}$$

Therefore, the Riccati equation is given by:

$$P\overline{A} + \overline{A}^T P - \left( P\overline{B} + R_{12} \right) R^{-1} \left( \overline{B}^T P + R_{12}^T \right) + Q = 0 \tag{52}$$

The aforementioned equation used C#'s Math.NET Numerics function to solve the Riccati equation. The known values $\overline{A}, \overline{B}, Q, R, R_{12}$ were input to obtain $P$.

After $P(t)$ is obtained, substituting Equation (51) into (49) yields the optimal controller:

$$U = -K_1 X - K_2 Z$$
$$= -R^{-1} B^T P(t) \begin{bmatrix} X \\ Z \end{bmatrix} - R^{-1} \begin{bmatrix} X & Z \end{bmatrix} R_{12} \tag{53}$$

Substituting Equation (53) into Equation (36), we can derive:

$$\dot{X} = (A - BK_1)X - BK_2 C_n^{-1} \widetilde{Y} \tag{54}$$

$$Y = CX + \left( -K_1 X - K_2 C_n^{-1} \widetilde{Y} \right) = (C - K_1)X - \left( K_2 C_n^{-1} \right) \widetilde{Y} \tag{55}$$

Thus, substituting the integral of the Equation (55) into (53), we can obtain:

$$U = -K_1 X(s) - K_2 C_n \widetilde{Y}^{-1}(s)$$
$$= \left\{ K_1 [SI - (A - BK_1)]^{-1} BK_2 C_n^{-1} - K_2 C_n^{-1} \right\} \widetilde{Y}(s)$$
$$= \begin{bmatrix} T_{11} & T_{12} \\ T_{21} & T_{22} \end{bmatrix} \widetilde{Y}(s) \tag{56}$$

Substituting the relations of $U = \begin{bmatrix} a_x^B & \dot{\theta}_s \end{bmatrix}^T$ and $\widetilde{Y} = \begin{bmatrix} f_{px} & \dot{\theta}_p \end{bmatrix}^T$ into the Equation (59), the equation is expressed as shown in Figure 5.

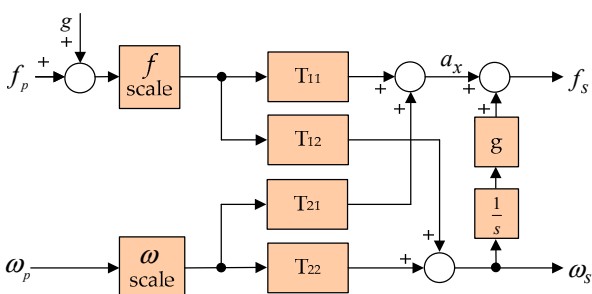

**Figure 5.** The block diagram of optimal control algorithm.

## 4. Simulation Results and Discussion

Figure 6 shows the relationship between the DSP (TMS-320F-28377D) and the PC. The type and parameters of the two motion-cueing algorithms are selected by the computer. After starting execution, the DSP opens the serial communication interface, transmits the input signals of specific force and angular velocity through the RS422 communication interface, and executes the motion-cueing algorithm by the interruption of the DSP. The DSP then sends the calculation results, displacement, angle, and acceleration to the PC so that the result can be drawn as a curve inside the PC. The interruption period of the DSP is around 1 ms. The average calculation time of the classical washout and the optimal control algorithms are around 800 and 136 μs, respectively. Although the classical washout algorithm is a simple and fast motion-cueing design method, considering the large number of filters, the design and implementation of the digital control system is very complicated, and its computation time is long compared with the optimal control algorithm. A user interface of the PC is established by the C# software (Figure 7a) and the HIL test bench by using an online simulation. With the HIL, we can identify and resolve problems earlier in the development cycle, as shown in Figure 7b. The HIL transmits and receives data between the PC and the control board through the RS422 communication interface which should be provided to enable modifications when required.

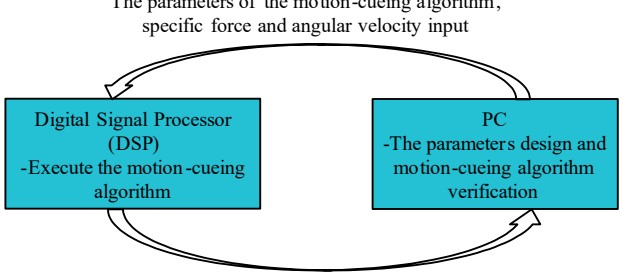

**Figure 6.** The relationship between the DSP and PC.

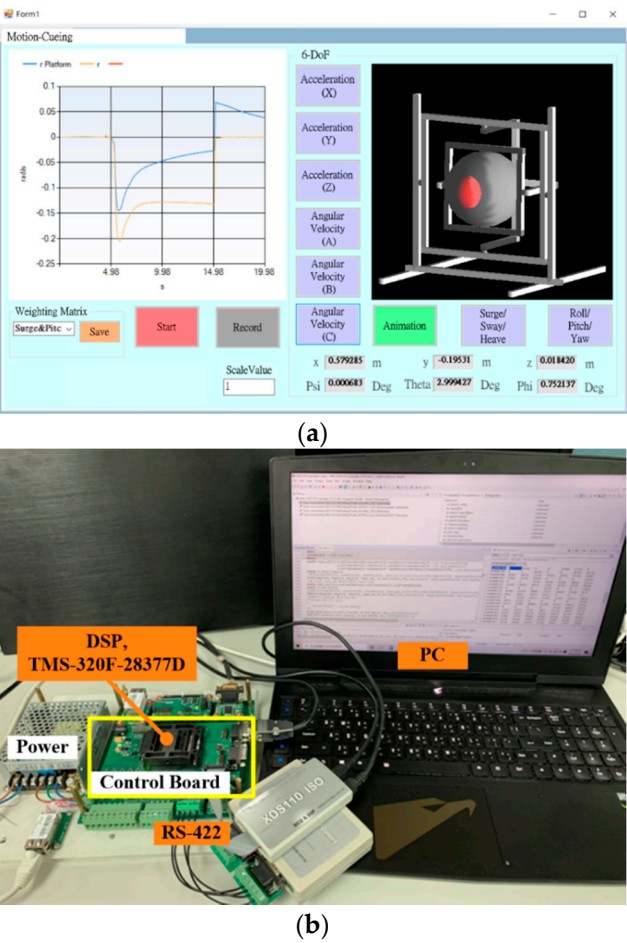

**Figure 7.** Hardware-in-the-loop (HIL) simulation platform: (**a**) user interface; (**b**) test bench.

Figure 8a shows the classical washout algorithm and the flowchart of the interrupt service routing. This method is used to process an input to produce movement as well as keep the motion within the workspace of the system. The flowchart of the interrupt service routing for the optimal control algorithm is shown in Figure 8b, and a comparison of the two motion-cueing algorithms is described in Section 4.3. The basic calculating principle is mentioned in Section 3. Finally, the output of the algorithms is obtained through a simple integral operation to obtain the displacement and angle.

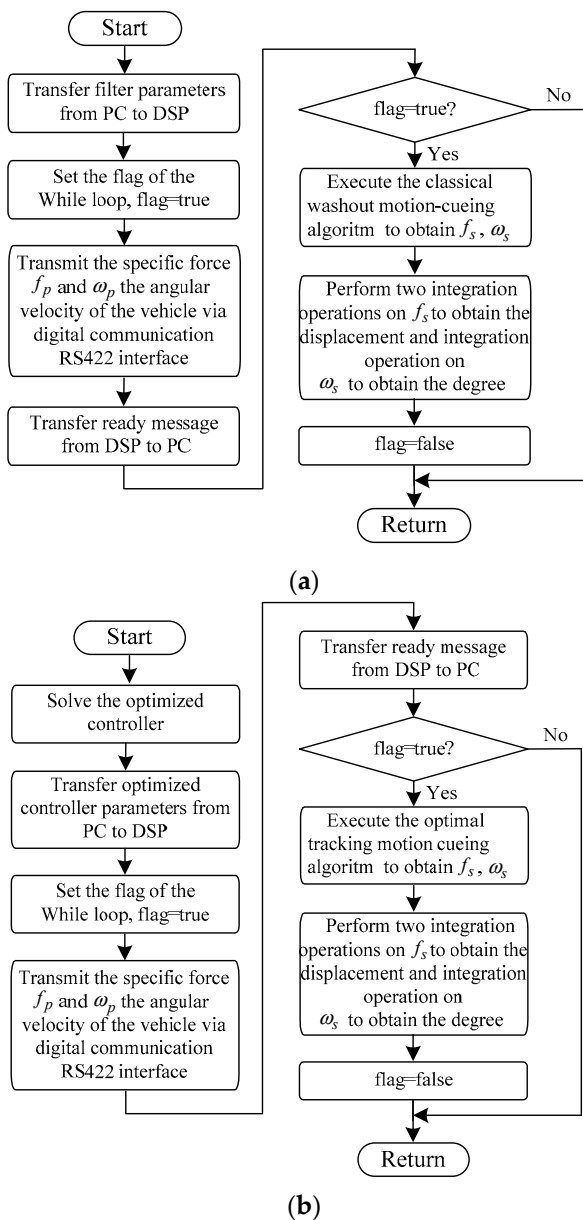

**Figure 8.** The flowchart of the DSP-based motion-cueing algorithms: (**a**) Classical washout algorithm; (**b**) Optimal control algorithm.

The simulation analysis elucidated the characteristics and performance of the optimal control algorithm. In the simulation, let $A_n = diag\{-\beta_1, -\beta_2\}$, $Q_1 = diag\{q_1, q_2\}$, $Q_2 = diag\{q_3, q_4, q_5\}$, and $R_1 = diag\{r_1, r_2\}$. The symbols $q_1$ and $q_2$ denote the penalties for specific force and errors, respectively, with a larger penalty yielding a more favorable tracking performance. The symbols $q_3$, $q_4$, and $q_5$ denote the penalties for displacement, motion velocity, and angle of the platform, respectively, with a greater penalty yielding a smaller platform displacement. The symbols $r_1$ and $r_2$ denote the penalties for the acceleration and platform angular velocity of the platform, respectively, with a greater penalty yielding a smaller acceleration and angular velocity generated by the platform, which in turn produces a smaller platform displacement that compromises tracking performance. The influences of the specific force analysis and angular velocity analysis of different parameters are shown in Figures 9–23 and Figures 24–26, respectively. The quantitative results are summarized in Tables 1 and 2.

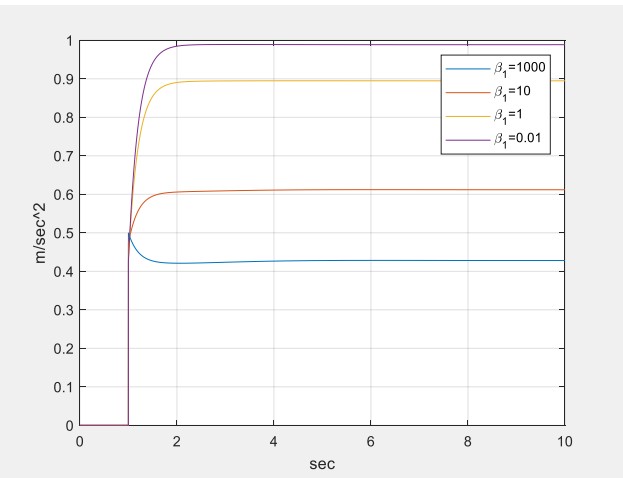

**Figure 9.** Specific force, $f_{sx}^{B}$, generated by the simulator.

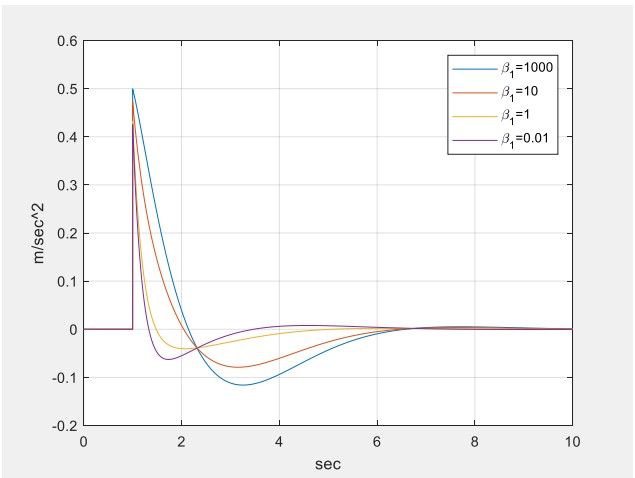

**Figure 10.** Acceleration, $a_x$, of the simulator (output of $T_{11}$).

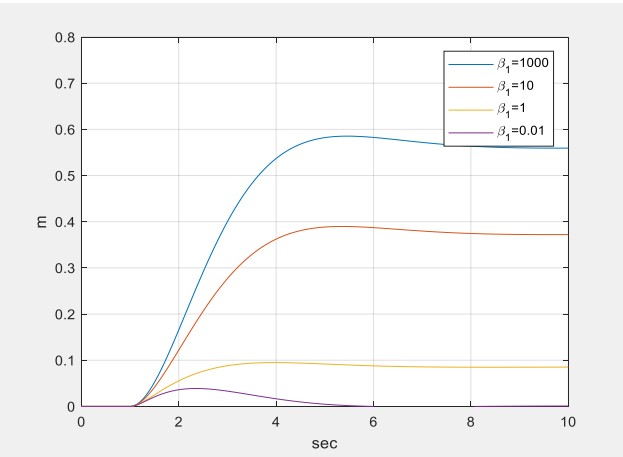

**Figure 11.** Displacement, $x_1$, of the simulator.

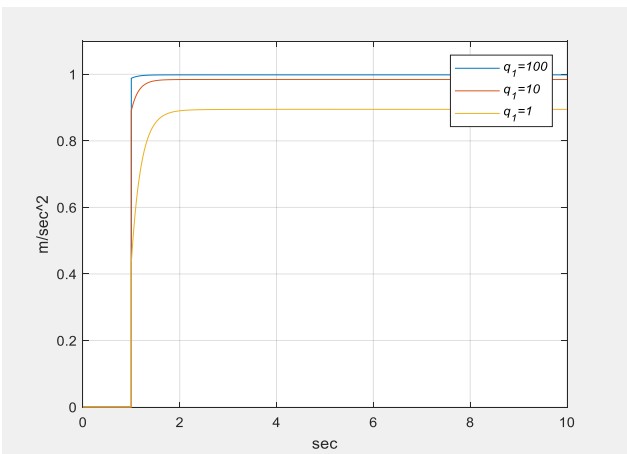

**Figure 12.** Effect of $q_1$ on specific force $f_{sx}^B$ generated by the simulator ($\beta_1 = 1$).

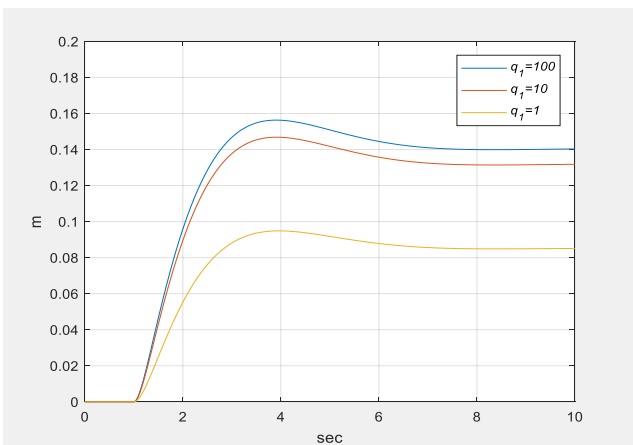

**Figure 13.** Effect of $q_1$ on the displacement generated by the simulator ($\beta_1 = 1$).

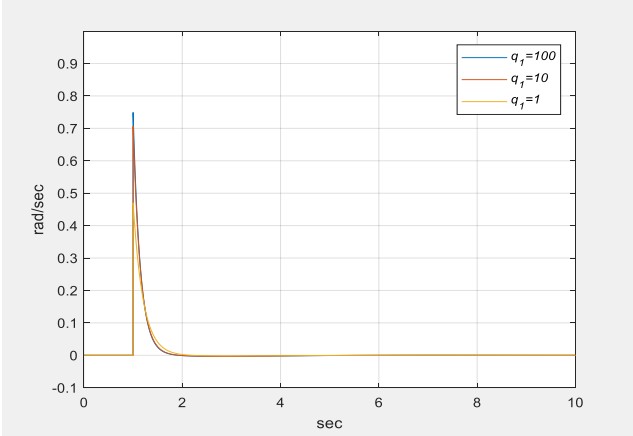

**Figure 14.** Effect of $q_1$ on the angular velocity $\dot{\theta}_s$.

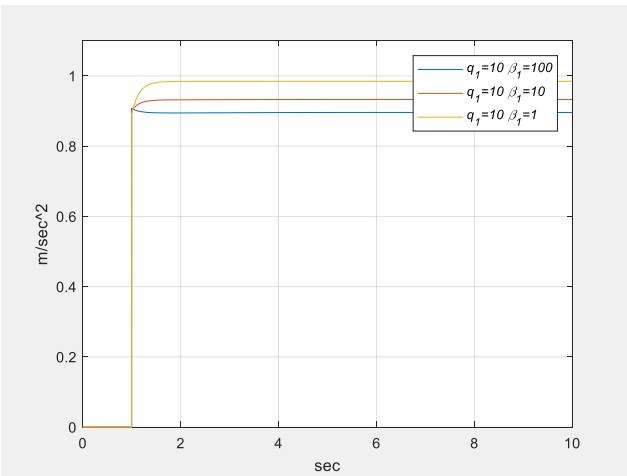

**Figure 15.** Verification of the influence of specific force tracking on angular velocity $\dot{\theta}_s$ and the influence of $\beta_1$ on specific force $f_{sx}^B$.

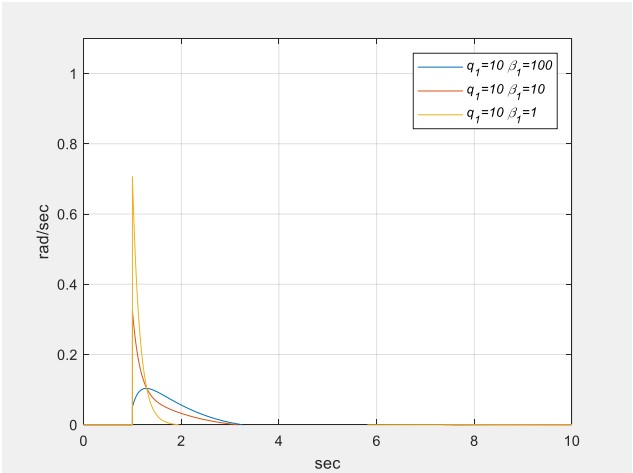

**Figure 16.** Verification of the influence of specific force tracking on angular velocity $\dot{\theta}_s$ and the influence of $\beta_1$ on $\dot{\theta}_s$.

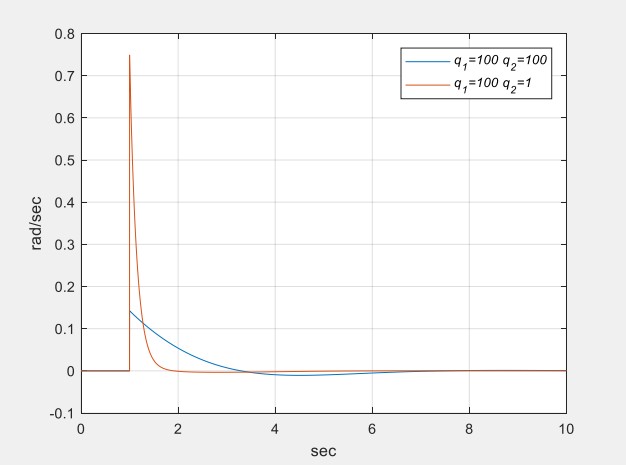

**Figure 17.** Improvement of the influence of specific force tracking on angular velocity $\dot{\theta}_s$ and the influence of $q_2$ on specific force $\dot{\theta}_s$.

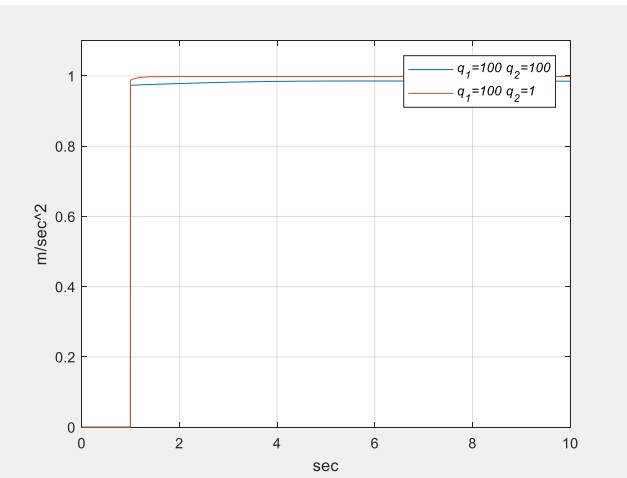

**Figure 18.** Improvement of the influence of specific force tracking on angular velocity $\dot{\theta}_s$ and the influence of $q_2$ on specific force $f_{sx}^B$.

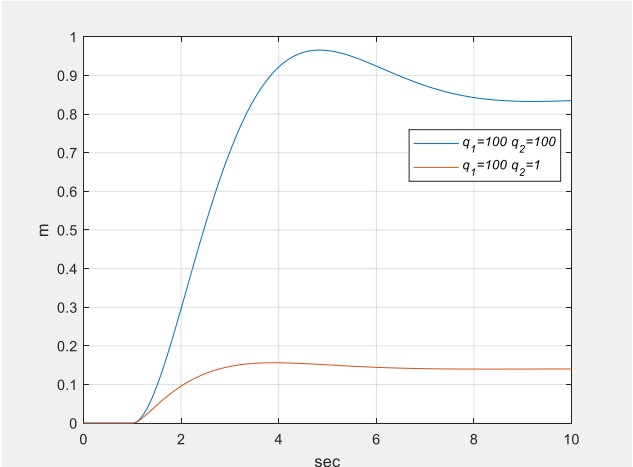

**Figure 19.** Improvement of the influence of specific force tracking on angular velocity $\dot{\theta}_s$ and the influence of $q_2$ on displacement.

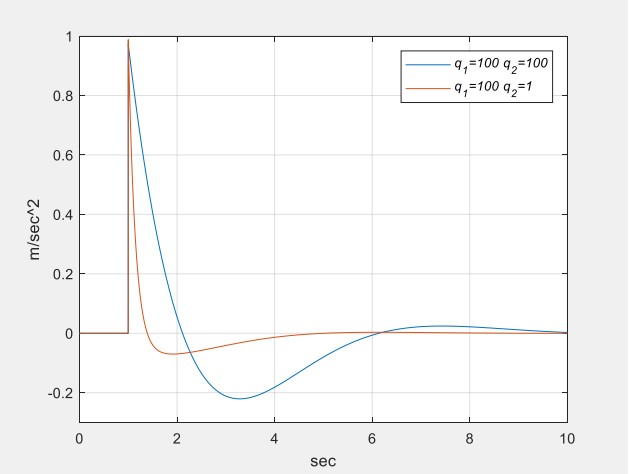

**Figure 20.** Improvement of the influence of specific force tracking on angular velocity $\dot{\theta}_s$ and the influence of $q_2$ on acceleration $a_x$.

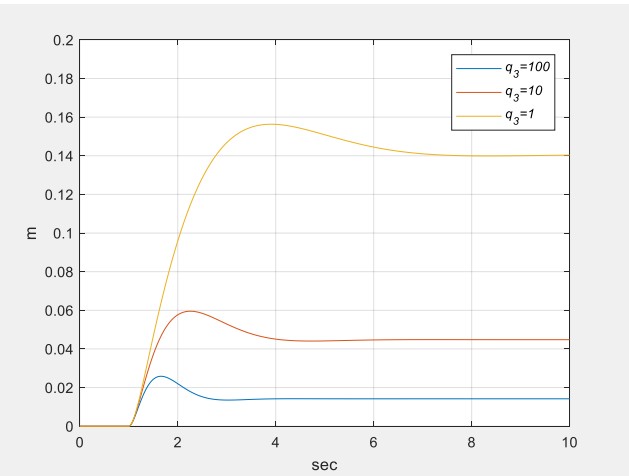

**Figure 21.** Influence of $q_3$ on displacement $q_1 = 100$, and $\beta_1 = 1$.

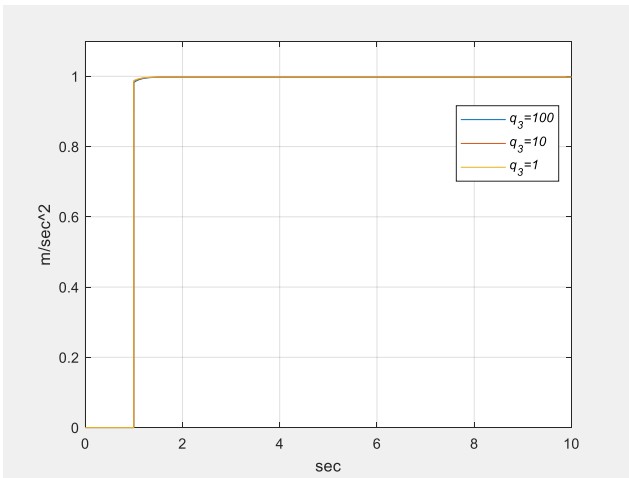

**Figure 22.** Influence of $q_3$ on specific force, $q_1 = 100$, and $\beta_1 = 1$.

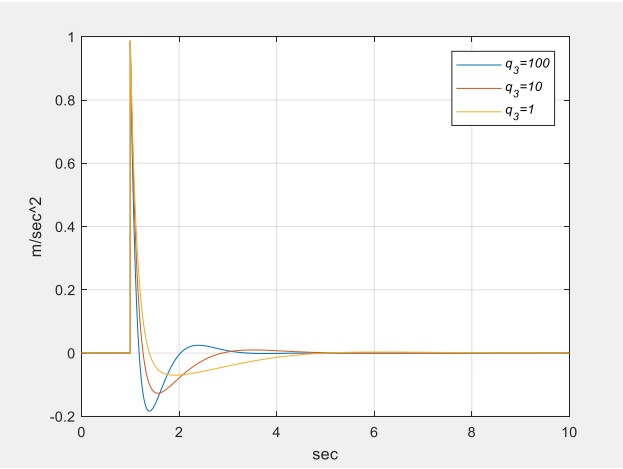

**Figure 23.** Influence of $q_3$ on acceleration $a_x$, $q_1 = 100$, and $\beta_1 = 1$.

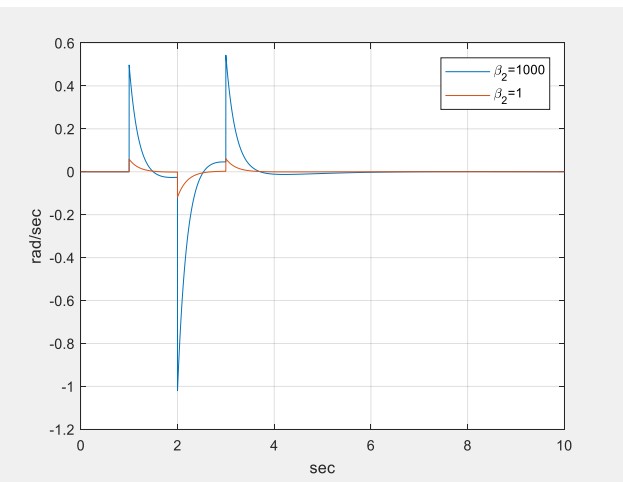

**Figure 24.** Influence of $\beta_2$ on angular velocity.

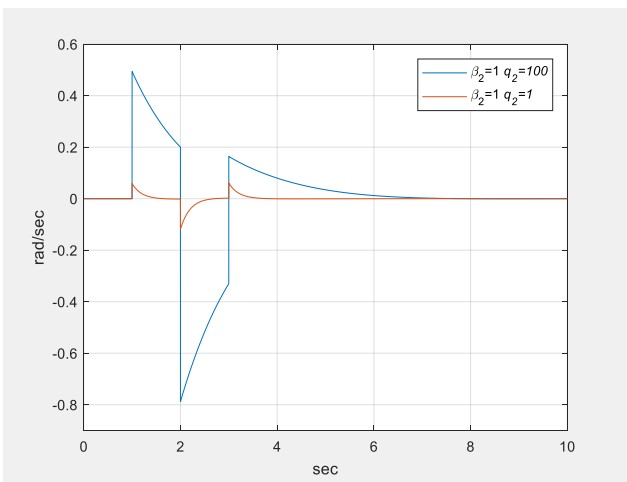

**Figure 25.** Influence of $q_2$ on angular velocity.

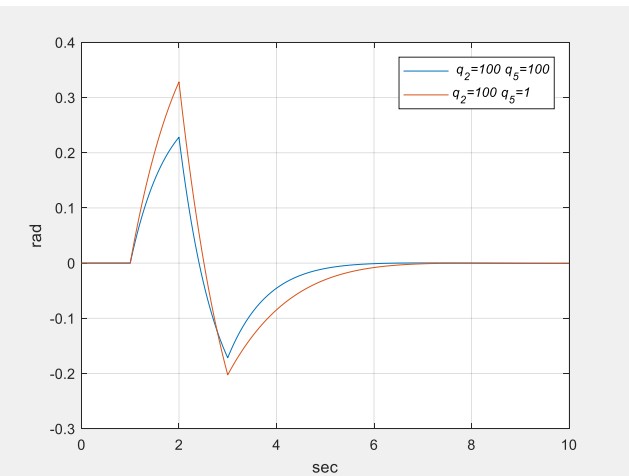

**Figure 26.** Influence of $q_5$ on the angle.

**Table 1.** The quantitative comparison of specific force.

| Input Command | Parameters | Figure | Specifications | |
|---|---|---|---|---|
| | | | Settling Time (s) | Steady State Value (m/s²) |
| Step-input (m/s²) 0, t < 1 +1, t > 1 | $\beta_2 = 1$, $q_2 = 1$, $q_3 = 1$, $q_4 = 0$, $q_5 = 1$, $r_1 = 1$, $r_2 = 1$ | Figure 9 | | |
| | $\beta_1 = 1000, q_1 = 1$ | | 2.9 | 0.42 |
| | $\beta_1 = 10, q_1 = 1$ | | 2 | 0.613 |
| | $\beta_1 = 1, q_1 = 1$ | | 1.5 | 0.893 |
| | $\beta_1 = 0.01, q_1 = 1$ | | 1.2 | 0.987 |
| | $\beta_1 = 1, q_1 = 100$ | Figure 12 | 0.3 | 0.999 |
| | $\beta_1 = 1, q_1 = 10$ | | 0.6 | 0.989 |
| | $\beta_1 = 1, q_1 = 1$ | | 1.5 | 0.893 |
| | $\beta_1 = 100, q_1 = 10$ | Figure 15 | 0.4 | 0.895 |
| | $\beta_1 = 10, q_1 = 10$ | | 0.5 | 0.93 |
| | $\beta_1 = 1, q_1 = 10$ | | 0.6 | 0.989 |

**Table 2.** The quantitative comparison of angular velocity.

| Input Command | Parameters | Figure | Specifications | |
|---|---|---|---|---|
| | | | Settling Time (s) | Peak to Peak Value |
| Step-input (rad/s) +1, 1 ≤ t ≤ 2 −1, 2 ≤ t ≤ 3 0, elswhere | $\beta_1 = 1$, $q_1 = 1$, $q_3 = 1$, $q_4 = 0$, $r_1 = 1$, $r_2 = 1$ | | | |
| | $\beta_2 = 1000, q_2 = 1, q_5 = 1$ | Figure 24 | 4.6 | 1.53 (rad/s) |
| | $\beta_2 = 1, q_2 = 1, q_5 = 1$ | | 2.7 | 0.15 (rad/s) |
| | $\beta_2 = 1, q_2 = 100, q_5 = 1$ | Figure 25 | 6 | 1.25 (rad/s) |
| | $\beta_2 = 1, q_2 = 1, q_5 = 1$ | | 2.6 | 0.17 (rad/s) |
| | $\beta_2 = 1, q_2 = 100, q_5 = 100$ | Figure 26 | 4.8 | 0.39 (rad) |
| | $\beta_2 = 1, q_2 = 100, q_5 = 1$ | | 5.3 | 0.53 (rad) |

*4.1. Specific Force Analysis*

4.1.1. Influence of $\beta_1$

In the simulation, where $\beta_2 = 1$, $q_1 = q_2 = 1$, $q_3 = q_5 = 1$, $q_4 = 0$, and $r_1 = r_2 = 1$, the simulations of $\beta_1$ at 1000, 10, 1, and 0.01 are compared. Figure 9 shows that at a higher $\beta_1$, the sustained cue was weak. However, $f_{sx}^B$ increased to a relatively high value at 1 s, which is less evident. Figure 10 clearly shows that the acceleration $a_x^B$ rapidly increased to the high-frequency gain of $T_{11}$ at 1 s. This was because $\beta_1$ represents the bandwidth of the signal to be tracked. A greater $\beta_1$ value indicates that a signal in a relatively wide frequency domain needs to be tracked and that the tracking capacity is dispersed in a relatively wide frequency domain. By contrast, smaller $\beta_1$ values indicate that a relatively low-frequency signal must be tracked and that the tracking capacity is focused on low-frequency signals. Therefore, a greater $\beta_1$ value yielded a weak sustained cue performance but a more favorable instant rise capacity at 1 s, which represented high-frequency performance. Figure 11 shows the simulator displacement $x_1$, as a greater $\beta_1$ required the tracking of signals in a wider frequency domain.

4.1.2. Influence of $q_1$

Figure 12 shows that a greater $q_1$ yielded more favorable tracking performance, and Figure 13 shows the influence of $q_1$ on displacement. However, a favorable specific force would influence the angular velocity. According to Figure 14, a greater $q_1$ produced a greater angular velocity, which increased to the high-frequency gain of $T_{12}$. This is mainly because the platform had to rapidly generate the angle required by the sustained cue for efficient specific force tracking. This phenomenon can be seen in Figures 15 and 16. The previous section illustrated the effect of $\beta_1$ on high- and low-frequency tracking performance; thus, $\beta_1$ was adjusted. Figure 15 shows that the sustained cue decreased (the

tracking performance of sustained cue declined) when $\beta_1$ increased. Moreover, the angular velocity $\dot{\theta}_s$ in Figure 16 decreased (the effect on the angular velocity cue was small).

### 4.1.3. Influence of $q_2$

The influence on angular velocity can be improved by enhancing the tracking performance of angular velocity, that is, by increasing $q_2$. Figure 17 shows that when $q_1 = 100$ (favorable specific force performance), increasing $q_2$ reduced the effect of specific force tracking on angular velocity. The classical washout algorithm relies on the angular velocity limiter to limit the effect of specific force on angular velocity. Figure 18 shows that increasing $q_2$ did not have a considerable impact on specific force tracking. However, Figure 19 shows that increasing $q_2$ simultaneously also increased the displacement. This is because, to attain favorable specific force performance while limiting the impact of angular velocity, more low-frequency acceleration was required for compensation, as shown in Figure 20, wherein the acceleration gradually became zero after $q_2$ was increased, indicating the presence of more low-frequency acceleration.

### 4.1.4. Influence of $q_3$

Here, the influence of $q_3$ (displacement penalty term of cost function) is investigated. In the simulation, let $q_1 = 100$, $\beta_2 = 1$. Figure 21 shows that a greater $q_3$ led to a smaller displacement. The influence of $q_3$ on displacement should also affect acceleration and the specific force at the initial stage. Figure 22 shows that an increase in $q_3$ slightly reduced the initial acceleration. Figure 23 also reveals that the acceleration became zero faster when $q_3$ increased, that is, $T_{11}$ filtered out more low-frequency signals.

### 4.2. Angular Velocity Analysis

The difference between the angular velocity and specific force is that specific force can generate a low-frequency motion by tilting at an angle, whereas angular velocity does not exhibit this mechanism and, therefore, sacrifices the low-frequency angular velocity that causes an excessive angle.

### 4.2.1. Influence of $\beta_2$

Figure 24 shows that a greater $\beta_2$ yielded a higher simulator movement angle. In the simulation, $\beta_1 = 1$, $q_1 = q_2 = 1$, $q_3 = q_5 = 1$, $q_4 = 0$, and $r_1 = r_2 = 1$. The $\beta_2$ primarily affected the high-frequency performance.

### 4.2.2. Influence of $q_2$

Figure 25 shows the influence of $q_2$ on angular velocity. The $q_2$ affected the overall tracking performance, and a more favorable performance generated a greater angular displacement.

### 4.2.3. Influence of $q_5$

Figure 26 shows the effect of $q_5$ on the angle. The symbol $q_5$ denotes the penalty for the angle, and a greater $q_5$ yielded a smaller angle, which in turn compromised the tracking performance of angular velocity.

### 4.3. Comparison of the Two Motion-Cueing Algorithms

The characteristics and performance of the optimal control motion-cueing algorithm were simulated and analyzed. The detailed qualitative comparison is shown in Table 3. Figures 2 and 5 show the block diagrams for the classical washout algorithm and optimal control algorithm, respectively. According to the figure, the classical washout algorithm was mainly composed of two high-pass filters and one low-pass filter, whereas the optimal control algorithm generated acceleration and angular velocity using four transfer functions (63).

**Table 3.** The qualitative comparison of different algorithms.

|  | Classical Washout Algorithm | Optimal Control Algorithm |
|---|---|---|
| Type | Filter-based | Optimization-based |
| Real-time capable | High | Medium |
| Scalability | High | High |
| Implementation complexity | High | Medium |
| Accounting for simulator limits | Through manual tuning | Through cost function optimization |
| Computation time | 800 μs | 136 μs |

### 4.3.1. Comparison of Origin Drift

We performed a simulation analysis to understand the characteristics and performance of the optimal control algorithm. Next, we let $\beta_1 = \beta_2 = 1$, $q_1 = q_2 = 1$, $q_3 = q_5 = 1$, $q_4 = 0$, and $r_1 = r_2 = 1$. In the simulation, because the classical washout algorithm without using the feedback control, the optimal control algorithm was used as a closed-loop control. Therefore, when the simulator drifted from the origin, the optimal control algorithm could facilitate the simulator's return to the origin, as shown in Figure 27a. If the initial position of drifting away from the origin in Figure 27b was assumed to be set at 1 m when the input specific force was 0, then the algorithm would return to the origin 10 s later.

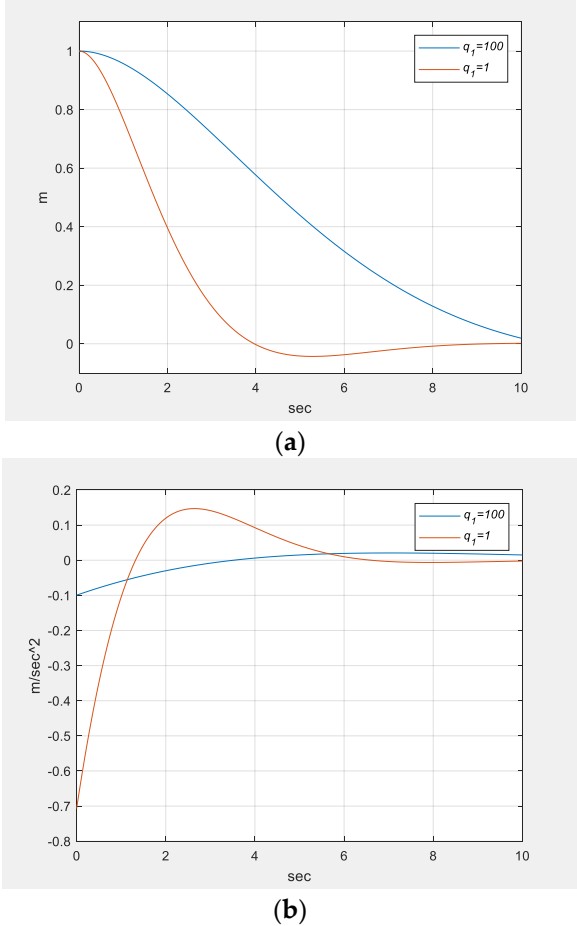

(a)

(b)

**Figure 27.** Optimal control when input specific force is 0: (**a**) Displacement result; (**b**) Specific force result.

### 4.3.2. Comparison of the Control Architecture

The $T_{11}$, $T_{12}$, $T_{21}$, and $T_{22}$ transfer functions corresponding to Equation (56) can be obtained, and are given as:

$$T_{11} = \frac{0.43247s^2(s + 0.9714)}{(s + 4.953)(s^2 + 1.403s + 0.9945)} \tag{57}$$

$$T_{12} = \frac{0.47016s(s^2 + 1.441s + 0.9565)}{(s + 4.953)(s^2 + 1.403s + 0.9945)} \tag{58}$$

$$T_{21} = \frac{0.027609s^2(s - 3.387)}{(s + 4.953)(s^2 + 1.403s + 0.9945)} \tag{59}$$

$$T_{22} = \frac{0.058597s(s^2 + 1.132s + 0.7647)}{(s + 4.953)(s^2 + 1.403s + 0.9945)} \tag{60}$$

The aforementioned equation shows that $T_{11}$ is a high-pass filter, $\frac{1}{s}T_{12}$ is a low-pass filter, $T_{21}$ is a high-pass filter, and $T_{22}$ is a high-pass filter. The optimal control motion-cueing algorithm was selected by the cost function of Equation (50) to obtain solution $P$ of the Riccati equation to acquire the controller parameters $K_1$ and $K_2$. Figures 2 and 5 show that the optimal control motion-cueing algorithm had an extra $T_{21}$ high-pass filter compared with the classical washout algorithm. Therefore, the rotational angular velocity $\dot{\theta}_p$ of the simulator cockpit is shown in Figure 28a. Figure 28b illustrates the effect of angle on the specific force with and without . Figure 28b indicates that the influence of the specific force in the blue line was smaller.

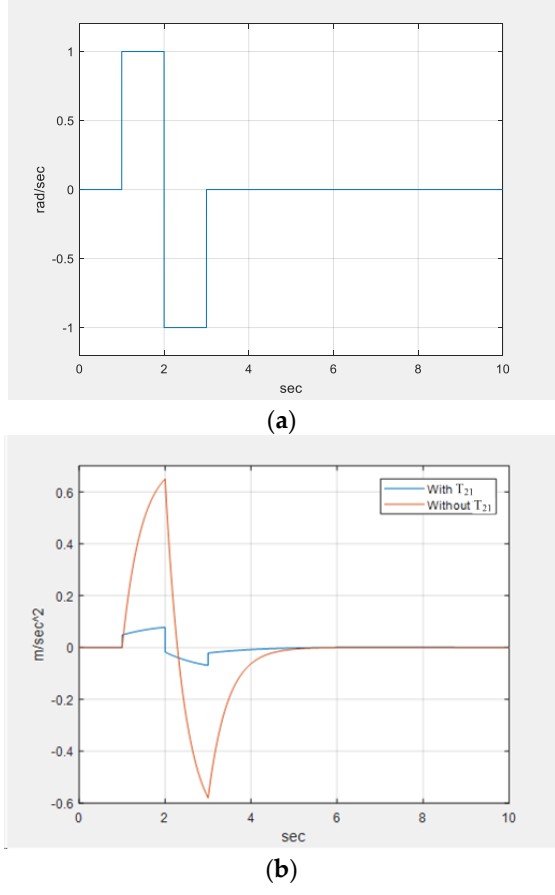

(a)

(b)

**Figure 28.** Comparison of motion-cueing algorithms: (**a**) input rotational angular velocity; (**b**) influence of $T_{21}$ angular motion on specific force.

## 5. Conclusions

The six-axis motion-cueing platform can produce different attitudes and movements. However, the movement space of the platform is limited. The classical washout algorithm transforms the corresponding rotation angle by using the tilt co-ordinates. The component of the acceleration of gravity deceived the training personnel. In addition, another algorithm designs the motion-cue algorithm based on the optimal control theory. Both are designed with embedded software to test the angular motion of linear motion, allowing the pilot to experience rotational angular velocity and specific force, and to compare the origin drift problem caused by the two algorithms working for a long time. Moreover, in the consistent real-time simulation and established kinematics model, an HIL test bench has been built based on a DSP-based motion-cueing algorithm. The aim of the study is to develop the algorithms implemented on a dedicated DSP of the HIL system and tested in the HIL environment. However, as further incorporation of the robust design of the platform is not this main topic of the study, but evaluation of the design should be considered in future work.

**Funding:** This research received no external funding.

**Conflicts of Interest:** The author declares no conflict of interest.

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
