# Peer review of "Design of a DSP-Based Motion-Cueing Algorithm Using the Kinematic Solution for the 6-DoF Motion Platform"

_aerospace, doi:10.3390/aerospace9040203_

Round 1
Reviewer 1 Report
Dear Authors,
The subject of the article as well as its content is interesting. However, I believe that the form of presentation should be slightly improved. For example, Figures 1a and 1b are crucial for further consideration. While 1a does not raise any doubts (it can only be slightly enlarged), the interpretation of 1b is problematic. It is worth presenting Figure 1b in a three-dimensional form, indicating and specifying on it (or in additional drawings) the components (linear, angular actuators, etc.)
Little space in the article is devoted to the description of the HIL simulation stand. There are no drawings, photos or detailed specifications related to the HIL environment in which the research was conducted.
Most of the presented mathematical relationships are very general and universal, while the obtained simulation results are the added and valuable value in this manuscript. I propose to extend the conclusions with qualitative and quantitative analysis of the obtained results.
Best regards,
Reviewer
Author Response
Reviewer#1, Concern # 1:
Comments and Suggestions for Authors
The subject of the article as well as its content is interesting. However, I believe that the form of presentation should be slightly improved. For example, Figures 1a and 1b are crucial for further consideration. While 1a does not raise any doubts (it can only be slightly enlarged), the interpretation of 1b is problematic. It is worth presenting Figure 1b in a three-dimensional form, indicating and specifying on it (or in additional drawings) the components (linear, angular actuators, etc.)
Authors’ response: Yes, we have revised this part.
Authors’ action: Please see lines 82-95 in page 2 and Figure 1(a) and (b).
Reviewer#1, Concern # 2:
Little space in the article is devoted to the description of the HIL simulation stand. There are no drawings, photos or detailed specifications related to the HIL environment in which the research was conducted.
Author response: We thank for the reviewer’s suggestion.
Author action: Please see from the line 310 in page 11 to line 326 in page 12 and Figure 7(a) and (b).
Reviewer#1, Concern # 3:
Most of the presented mathematical relationships are very general and universal, while the obtained simulation results are the added and valuable value in this manuscript. I propose to extend the conclusions with qualitative and quantitative analysis of the obtained results.
Author response: Yes, we have revised.
Author action: Please see lines 357-358, Table 1-2 in page 14 and lines 468-469, Table 3 in page 22.

Reviewer 2 Report
Dear author,
thanks for your work. Motion cueing platforms have always attracted my attention, being useful in vehicle simulation for training and testing.
The paper is well structured, but I'd focus the attention to few things:
- English needs a review. There are several sentences that must be corrected and make the article difficult to read.
- the section about kinematics is extensive, but complete and, although sometimes seem obvious, it is useful to guide the reader to understand your scope.
- The section about results need some work: seeing figures, it seems that authors tried their algorithms in a simulation without uncertainties and sensor noise. In my opinion this is a fundamental step of simulation results, in order to emulate the reality and test the algorithms in a real scenario. Being a non-robust design based on a kinematic model, it would be a good add-on to prove the effectiveness.
I hope my comment will be useful to improve your work.
Kind regards
Author Response
Reviewer#2, Concern # 1:
Comments and Suggestions for Authors
Dear author, thanks for your work. Motion cueing platforms have always attracted my attention, being useful in vehicle simulation for training and testing.
Authors’ response: We appreciate your positive comments on this manuscript.
Reviewer#2, Concern # 2:
The paper is well structured, but I'd focus the attention to few things:
English needs a review. There are several sentences that must be corrected and make the article difficult to read.
Authors’ response: We thank for the reviewer’s suggestion.
Authors’ action: We do our best to revise the presentation and English in the whole manuscript to make it more readable. Thank you very much.
The section about kinematics is extensive, but complete and, although sometimes seem obvious, it is useful to guide the reader to understand your scope.
Author response: Thanks for your suggestion.
Author action: We have revised the whole manuscript to make it more readable. Thanks for your valuable comments.
The section about results need some work: seeing figures, it seems that authors tried their algorithms in a simulation without uncertainties and sensor noise. In my opinion this is a fundamental step of simulation results, in order to emulate the reality and test the algorithms in a real scenario. Being a non-robust design based on a kinematic model, it would be a good add-on to prove the effectiveness.
Authors’ response: Yes, we thank for the reviewer’s suggestion.
Authors’ action: Please see from the line 310 in page 11 to line 326 in page 12, Figure 7(a) and (b), and lines 521-527 in page 24.
I hope my comment will be useful to improve your work.
Authors’ response: Yes, we thank for the reviewer’s valuable comment.
Authors’ action: We have made the required revisions. Thank you very much.

Round 2
Reviewer 1 Report
The manuscript has been sufficiently improved. Thank you!Reviewer 2 Report
Dear Authors,
thank for your revision. Now the paper appears clearer and results are more complete.
Although I consider this kind of paper a sort of tutorial rather than a scientific article, I think it is now ready for publication.
Kind regards